# Effect of Heart Rate Reserve on Prefrontal Cortical Activation While Dual-Task Walking in Older Adults

**DOI:** 10.3390/ijerph19010047

**Published:** 2021-12-21

**Authors:** Alka Bishnoi, Gioella N. Chaparro, Manuel E. Hernandez

**Affiliations:** 1Department of Kinesiology and Community Health, College of Applied Health Science, University of Illinois at Urbana-Champaign, Urbana, IL 61801, USA; abishn2@illinois.edu; 2Department of Kinesiology, California State University, Dominguez Hills, Carson, CA 90747, USA; gchaparro@csudh.edu

**Keywords:** neuroimaging, cognition, gait

## Abstract

Hypertension is considered a risk factor for cardiovascular health and non-amnestic cognitive impairment in older adults. While heart rate reserve (HRR) has been shown to be a risk factor for hypertension, how impaired HRR in older adults can lead to cognitive impairment is still unclear. The objective of this study was to examine the effects of HRR on prefrontal cortical (PFC) activation under varying dual-task demands in older adults. Twenty-eight older adults (50–82 years of age) were included in this study and divided into higher (*n* = 14) and lower (*n* = 14) HRR groups. Participants engaged in the cognitive task which was the Modified Stroop Color Word Test (MSCWT) on a self-paced treadmill while walking. Participants with higher HRR demonstrated increased PFC activation in comparison to lower HRR, even after controlling for covariates in analysis. Furthermore, as cognitive task difficulty increased (from neutral to congruent to incongruent to switching), PFC activation increased. In addition, there was a significant interaction between tasks and HRR group, with older adults with higher HRR demonstrating increases in PFC activation, faster gait speed, and increased accuracy, relative to those with lower HRR, when going from neutral to switching tasks. These results provide evidence of a relationship between HRR and prefrontal cortical activation and cognitive and physical performance, suggesting that HRR may serve as a biomarker for cognitive health of an older adult with or without cardiovascular risk.

## 1. Introduction

Approximately 32.2% of the adults in United States have been diagnosed with high blood pressure or hypertension [1]. Hypertension is the fifth leading cause of death in older adults [2]. Hypertension has been identified as a risk factor for mild cognitive impairment (MCI) and dementia in observational studies [3]. Hypertension is not only a risk factor for stroke and small vessel disease [4] but may also be responsible for a higher likelihood of cognitive impairment due to its role in cerebrovascular disease, subcortical white-matter lesions, blood–brain barrier dysfunction, or formation of free oxygen radicals [5]. Evidence suggests that chronic hypertension, especially high systolic blood pressure during midlife (40–65 years), is associated with an increased risk of cognitive decline and dementia in late adulthood [6] (Figure 1). One of the major risk factors of hypertension is lower heart rate reserve (HRR) [7], which is defined as a capacity difference between the maximum heart rate that an individual can achieve and their resting heart rate. Given that hypertension is a risk factor for non-amnestic cognitive impairments [8], the examination of older adults with associated risk factors such as impaired HRR may provide an important marker of executive function changes of at-risk individuals at an early stage.

HRR is also linked with chronotropic response, which is described as the heart’s ability to adequately respond to increased activity and is usually measured by a graded exercise stress test [9]. An increased chronotropic response is associated with a better cognitive performance and higher HRR [10]. Thus, as hypertension is linked with a lower HRR [7] which has also been directly related to a lower cognitive function [10], it is important to examine executive function processes in older adults with lower HRR. This can be achieved by the performance of concurrent cognitive tasks while walking, such as walking while talking (i.e., dual-task walking), as these tasks commonly involve executive function processes [11]. Executive function processes, predominantly activated in the prefrontal cortex, involves multiple subdomains such as inhibition, working memory, mental tracking, and planning, which are crucial for control of gait, particularly during dual-task walking in older adults [12,13]. Given that executive function is necessary to effectively perform dual-task walking, prefrontal cortical (PFC) activity may serve as a surrogate measure of executive function and attentional demands while walking in older adults [13] with lower HRR.

Through the use of neuroimaging methods such as functional near-infrared spectroscopy (fNIRS), PFC activity can be measured using the relative changes in oxygenated (HbO_2_) and deoxygenated (Hb) hemoglobin levels and, thus, provides a crucial measure of cognitive control of gait in older adults [14,15,16,17]. fNIRS has been found to be a reliable measure of hemodynamic responses in the PFC while walking [14,15,16,17]. Previous research has found a positive relationship between PFC activation patterns and aerobic fitness in cognitively demanding tasks, such that fitter individuals exhibit increased PFC recruitment as task difficulty increases [18,19]. Older adults have typically demonstrated increased PFC activation when dual-task walking relative to baseline conditions [12,16,20], but not always [21,22]. Age-related comparisons have been inconclusive as healthy older adults have shown decreased [12], increased [20], or unchanged [21,22] PFC activation levels during dual-task walking conditions, compared to young adults, which may be due to specific task demands or choice of baseline. Thus, while we seek to examine the effect of HRR on PFC activation levels in older adults, it will be important to account for the age-related and aerobic capacity changes in our cohort to further contribute to this growing field.

The Modified Stroop Color Word Test (MSCWT) can successfully encompass a wide array of executive function processes such as inhibition, set switching, flexibility, and working memory [23]. Chaparro et al. [24] has shown the effect of MSCWT on cognitive performance in older adults while standing or walking and found lower cognitive performance as evaluated by higher Stroop interference, particularly during the switching block. Furthermore, increased age and decreased aerobic capacity have been associated with decreased cognitive performance during the switching block while walking [24]. Thus, the use of the MSCWT while walking may provide a sensitive measure of cognitive function changes in persons with impaired HRR.

The objective of this pilot study is to examine the effect of HRR on PFC activation while doing the MSCWT while walking, which may provide an analogous measure to walking while shopping or switching thinking tasks while walking in the community. Given the sensitivity of the MSCWT while walking to detect age-related declines in the ability to change task focus and the potential pathway toward cognitive decline due to impaired HRR, our hypothesis is that older adults with higher HRR, in comparison to lower HRR, will demonstrate higher PFC activation levels and better cognitive and physical performance while performing the MSCWT during walking, particularly as the difficulty of the task increases, consistent with prior observations in fitter individuals [18,19]. Thus, this study will provide insight into how HRR may impact PFC activity as the difficulty of the concurrent cognitive task increases and identify potential benefits of dual-task assessments in persons with impaired HRR.

## 2. Materials and Methods

### 2.1. Participants

Twenty-eight participants (71 ± 17.4 years of age (mean ± SD), range 50–82 years of age, 18 females) were recruited from the local community for this two-session cross-sectional laboratory study. Inclusion criteria consisted of no report of any neurological, orthopedic, or cardiovascular conditions such as heart failure, coronary artery disease, arrhythmias, and hypertension, no physical disabilities, and being able to walk independently without an assistive device. Exclusion criteria consisted of presence of dementia based on a score <19 on the Telephone Interview for Cognitive Status [31], depression based on a score >10/30 on the short Geriatric Depression Scale [32], and red–green color blindness as measured with the Ishihara 38 Plates CVD test on Colblindor [33]. All participants signed a written consent form approved by the local institutional review board prior to testing.

### 2.2. Procedures

All participants made two separate visits to the Mobility and Fall Prevention Research Lab at the University of Illinois at Urbana-Champaign. Day 1 was dedicated to obtaining informed consent forms, performing physical and cognitive assessments, receiving treadmill training, and performing the Rockport 1 Mile Walk Test [34]. On Day 2, before the start of the dual-task paradigm, participants received instruction and training on the MSCWT (with the same number of trials for each block as the testing conditions). Training occurred in a seated position in front of a computer monitor. Prior to that, participants took part in the dual-task paradigm and completed questionnaires and surveys.

### 2.3. Physical and Cognitive Assessments

During their first visit, participants had their cognitive function assessed using the Repeated Battery for the Assessment of Neuropsychological Status (RBANS) [35] and their premorbid intellectual functioning assessed using the National Adult Reading Test (NART) [36]. Education was assessed as the number of years of higher education post high school. The Rockport 1 Mile Walk test was used to evaluate the aerobic fitness of participants, given its demonstrated validity and strong correlation with direct measures of VO_2_ max [37]. During this test, participants walked 1 mile (eight laps) in an indoor track as briskly as possible while wearing a heart rate monitor (T5H911 Timex digital fitness heart rate monitor). For assessing HRR, a research assistant recorded the baseline resting heart rate (beats per minute) and post heart rate at the completion of the 1 mile test. HRR was calculated by taking the difference between maximum heart rate, calculated using an age-predicted maximal heart rate formula (208–0.7 × age) [38], and resting heart rate [39]. Participants were separated into two groups of HRR using a cutoff of 76 to provide a bivariate split of participants: lower HRR (*n* = 14), higher HRR (*n* = 14) (Table 1). 

### 2.4. Dual-Task Paradigm

During their second visit, participants performed the MSCWT while standing and walking in a counterbalanced order. The MSCWT consisted of four separate blocks: neutral, congruent, incongruent, and switching [24,40] (Figure 2). Each block in the MSCWT consisted of 40 trials (except neutral for 20 trials). During the switching block, the task switched every 3–5 trials, and the colored stimuli were pseudorandomized. Each stimulus appeared on the screen in front of the self-paced instrumented treadmill (C-Mill, Motekforce Link, Culemborg, the Netherlands) for 2.5 s or until an actual verbal response was received from a participant (such as red, yellow, blue, and green). To limit the attentional demands on the participant, a trained research assistant registered all participants’ verbal responses by pressing the corresponding button on a game controller, similar to prior work [41]. If there was no response within the stimulus period, that trial was eliminated from analysis. Before the start of each trial, a fixation cross appeared for 1.5 s to allow participants to prepare to make a response. Blocks were arranged from the easiest to most difficult task (Figure 2). Participants completed the dual-task paradigm on the self-paced instrumented treadmill while wearing a harness. Response accuracy (%) and gait speeds (m/s) were collected during the dual-task walking trials for characterizing cognitive and physical performance. The response accuracy was collected through the E-Prime software package (E-prime, Psychology Software Tools, Inc., Sharpsburg, PA, USA). Accuracy was selected as the variable for the MSCWT due to the high variability encountered in encoding the verbal response to collect reaction time, arising from the difference in verbal response of the participant and registering of the response by the trained research assistant. Average gait speed in each walking trial was calculated using custom MATLAB scripts using spatiotemporal data collected using CueFors 2 software (Motekforce Link, Culemborg, the Netherlands). In this study, we focused on results from the dual-task walking conditions.

### 2.5. Functional Near-Infrared Spectroscopy

PFC activation levels were recorded with a commercial functional near-infrared spectroscopy (fNIRS) system (fNIR Imager 1200 model, fNIR devices LLC, Potomac, MD, USA). The fNIRS sensor headband consisted of 10 photodetectors and four LED light sources with a 2.5 cm source–detector separation distance that provided coverage of the forehead with 16 optodes and a 2 Hz sampling rate. The center of the headband sensor was placed on the central point of the forehead above the nasion (Fpz), in accordance with the 10/20 system used in electroencephalography. 

### 2.6. Data Analysis

Custom MATLAB scripts were used to process the fNIRS data. The data collected from each of the 16 fNIRS optodes under each of the experimental conditions were carefully inspected and removed from analysis if saturation or dark current conditions were identified by a trained researcher. Raw data were then low-pass filtered with a finite impulse response filter of cutoff frequency at 0.14 Hz to eliminate possible respiration, heart rate signals, and unwanted high-frequency noise [42]. Using the modified Beer–Lambert law, deoxygenated (Hb) and oxygenated (HbO_2_) hemoglobin levels for each optode were calculated from the artefact-removed raw intensity measurements at 730 and 850 nm [43]. The task-related changes were measured by averaging Hb and HbO_2_ levels during the dual-task walking trials and comparing it to baseline values while standing and counting silently just prior to each condition. Individual mean Hb and HbO_2_ data were extracted separately for each of the 16 optodes in each of the experimental conditions.

### 2.7. Statistical Analysis

Descriptive statistics for gender, age, aerobic capacity (VO_2_ max), education level, heart rate reserve, RBANS, NART, and Rockport gait speed are provided, with differences between HRR groups tested using either a chi-square test for gender or an independent *t*-test for continuous variables (age, VO_2_ max, RBANS, NART, education level, Rockport gait speed) (Table 1). We used linear mixed models to evaluate any significant differences between LHRR and HHRR as a two-level between-subject factor, dual-walking task (neutral vs. congruent vs. incongruent vs. switching) as a four-level repeated within-subject factor, and gait speed or accuracy rate as the dependent measure. The interaction term of HRR group by task was also included, and a random intercept was included in the model to allow the entry point to vary across individuals. 

A linear mixed effects model was also used to examine the effect of HRR group and task, as well as its interaction term, on PFC activation, as measured by mean HbO_2_ levels and Hb levels, while controlling for repeated measures across the 16 optodes in analysis. Random intercepts by individuals across optodes were included. Given the differences observed in age and aerobic capacity (Table 1) between HRR groups, these factors were added as covariates in each linear mixed effects model. Data were visually inspected for normality and homogeneity of variance, and the assumptions for the linear mixed effects models were met by all models except accuracy rate. For accuracy rate, a rankit transformation was used. For all statistical tests, R was used, and significance was set at *p* < 0.05. A priori sample size calculations based on earlier pilot data suggested a minimum of 26 participants needed in this study to achieve 0.85 power, assuming α = 0.05 and an effect size of 0.25.

For secondary analysis, we conducted a region-based fNIRS analysis to see differences in PFC activation among hemispheres in two groups. We conducted a linear mixed effects model analysis for PFC activation as measured by HbO_2_ and Hb levels, including main factors hemisphere (left vs. right), region (lateral vs. medial), cohort, and task while controlling for age and aerobic capacity. Random intercepts by individuals were included.

## 3. Results

The 28 older adults that participated in the study were divided into LHRR (*n* = 14) and HHRR (*n* = 14) (Table 1). We found that age was significantly higher in the LHRR group relative to the HHRR group (*p* = 0.0002), and VO_2_ max was lower in the LHRR group compared to HHRR group (*p* = 0.019). We found that overground walking speed as measured by the Rockport 1 mile test was significantly higher in the HHRR group relative to the LHRR group (*p* = 0.030). No statistically significant differences in gender, education, NART, or RBANS were found between groups.

### 3.1. Physical and Cognitive Performance

Statistically significant increased gait speed during the switching MSCWT task, relative to the neutral task, was found across both cohorts (switching task: β = 0.1136, SE = 0.04, *p* = 0.009, Table 2). Furthermore, a statistically significant interaction between cohort and task was found, such that the HHRR group walked faster than the LHRR group during congruent relative to neutral condition (HHRR cohort × congruent task: β = 0.1257, SE = 0.06, *p* = 0.04, Table 2). No other statistically significant effects on gait speed were found (*p* > 0.05).

A statistically significant decrease in accuracy rate was observed in incongruent and switching tasks, relative to neutral tasks across both cohorts (incongruent task: β = −1.01, SE = 0.15, *p* ≤ 0.001; switching task: β = −1.91, SE = 0.15, *p* ≤ 0.001). Furthermore, a statistically significant interaction between cohort and task was found, such that the HRRR group demonstrated a higher accuracy rate than LHRR during switching relative to neutral condition (HRRR cohort × switching task: β = 0.56, SE = 0.24, *p* = 0.02, Table 2). However, no other statistically significant differences were observed. 

### 3.2. Prefrontal Cortical Activation

HbO_2_ levels in older adults with higher versus lower HRR are depicted in Figure 3. Overall, HHRR demonstrated increased HbO_2_ levels in comparison to LHRR, particularly in more challenging MSCWT tasks, as observed by a statistically significant interaction between tasks and HRR group, while controlling for age and aerobic capacity differences (*p* < 0.05, Table 3). Furthermore, as task difficulty increased, HbO_2_ levels increased from neutral to congruent, neutral to incongruent, and neutral to switching tasks (*p* < 0.001, Table 3). Older adults with increased age and increased aerobic capacity were found to have increased HbO_2_ levels (*p* < 0.05). Channels 12 and 14 were found to demonstrate increased HbO_2_ levels compared to Channel 1 (*p* < 0.05). No other statistically significant differences were observed in HbO_2_ levels.

Overall, HHRR demonstrated a statistically significant decrease in deoxyhemoglobin (Hb) levels in comparison to LHRR, while controlling for age and aerobic capacity differences (*p* < 0.05, Table 4, Figure 3). A statistically significant interaction was observed between tasks and HRR group, with HHRR decreasing Hb levels, in comparison to LHRR, when going from neutral to incongruent tasks (*p* < 0.05, Table 4, Figure 3). Furthermore, as task difficulty increased, Hb levels decreased from neutral to congruent, neutral to incongruent, and neutral to switching tasks (*p* < 0.05, Table 4). Older adults with increased age were found to have increased Hb levels, while those with increased aerobic fitness were found to have decreased Hb levels (*p* < 0.05). Channels 8, 10, 12, and 14 were found to demonstrate increased Hb levels compared to Channel 1 (*p* < 0.05). No other statistically significant differences were observed in Hb levels. 

Furthermore, we performed regional fNIRS analysis to see if the PFC activation defined by HbO_2_ and Hb levels would show significant effects in hemisphere (left vs. right) and region (lateral vs. medial), while controlling for age and aerobic fitness differences. No significant hemisphere or region effects with HbO_2_ and Hb levels or two-way interaction effects between hemisphere, region, and groups in HbO_2_ and Hb levels were recorded. 

## 4. Discussion

This is the first study investigating the effects of HRR on PFC activation while dual-task walking. Older adults with HHRR, relative to those with LHRR, walked faster, performed an MSCWT more accurately while walking, and demonstrated higher PFC activation levels, particularly in more challenging dual-task walking conditions, consistent with the “supply and demand” framework [44] and an increased neural reserve in older adults with HHRR [45]. Partly confirming our hypothesis, we found that HHRR (in comparison to LHRR) demonstrated higher PFC activation levels while dual-task walking that were modulated by task difficulty. However, instead of PFC activation levels increasing as task difficulty increased, there were smaller increases in older adults with LHRR, consistent with an increased neural reserve in older adults with HHRR [45]. 

Consistent with prior work demonstrating increased PFC activation in older adults while dual-task walking [46,47], we observed PFC activation increases when going from a from an easier dual task to a more difficult dual task (neutral, congruent, and incongruent to switching walking tasks). Decreased accuracy rates were observed in both cohorts in incongruent and switching tasks, relative to neutral tasks, consistent with the increased difficulty observed during these more challenging conditions while walking [24]. Furthermore, the increase in gait speed observed across both cohorts when going from neutral to switching may be the result of the self-paced treadmill walking mode used in this study. Consistent with prior studies of dual-task gait when using a self-paced treadmill [40,48], the participant’s control of their comfortable pace may present a concurrent motor task for participants that is lower in priority. 

Consistent with an increased neural reserve [45], older adults with HHRR, relative to LHRR demonstrated increases in PFC activation levels across all dual-task walking conditions, as measured by relative changes in Hb levels. Furthermore, adults with LHRR were unable to recruit more of the frontal areas than adults with HHRR or achieve parity in performance as the demands of the task increased, particularly in switching tasks, consistent with prior findings of increased PFC activation in higher fit adults [18,19]. These decreases in activation in more cognitively demanding tasks may be partly explained by the “supply and demand” framework [46], which posits that age-related or disease-related structural declines may result in decreased availability of cognitive resources. Furthermore, according to the capacity-sharing theory [49], a capacity overload may be expected in adults with LHRR, leading to the use of a “posture first strategy”, which would prioritize the motor over the cognitive task [50], particularly as the difficulty of the cognitive task increased beyond the available capacity. Thus, given the differences observed in PFC activation in LHRR and HHRR groups across different dual-task walking conditions, the MSCWT may provide a sensitive cognitive test for monitoring cerebrovascular changes in older adults.

In contrast with recent work in less fit older adults [51] or adults with higher cardiovascular disease risk factors, such as hypertension or obesity [52], we did not find increased neural activation while dual-task walking in older adults with LHRR. However, these differences may be due to the use of different baselines in measurement of neural activation and differences in populations, as older adults with clinically diagnosed cardiovascular conditions were excluded in the present study. Within our cohort of older adults, increased aerobic capacity was consistently associated with increased PFC activation, whereas increased age demonstrated mixed findings, which further supports the importance of using both HbO_2_ and Hb levels in evaluating neural activation while walking, as recommended by recent consensus criteria for fNIRS data processing [17].

Consistent with the hemispheric asymmetry reduction in older adults theory [53], older adults across both cohorts did not demonstrate lateralized PFC activation, according to secondary regional fNIRS analysis. In older adults, bilateral activation of brain areas has been observed in the performance of cognitive tasks [54,55], consistent with our findings. However, future work should examine younger adults during similar dual-task walking paradigms to further evaluate this theory. 

From a clinical perspective, it is important to understand that, if an adult is going through impaired cardiovascular health, then there is a possibility of non-amnestic cognitive impairment which will affect their everyday instrumental activities and may lead to increased fall risk and mobility impairment [52]. Therefore, while assessing walking performance and fall risk in adults with impaired cardiovascular health, it seems important to also assess their cognitive demands through dual-task walking because greater cognitive demands would likely interfere with the motor performance. In addition to this, designing dual-task interventions in adults with impaired cardiovascular health may help in decreasing cognitive decline and inhibiting the cognitive impairment before it takes place [52]. 

Overall, fNIRS provides a valid and reliable measure of prefrontal activation in adults. As demonstrated by concurrent functional magnetic resonance imaging and fNIRS studies, fNIRS measures provide a valid measure of neural activation [56,57,58]. Furthermore, fNIRS measures have demonstrated good to moderate reliability in prior motor and walking tasks [59,60,61] and application in numerous dual-task walking conditions and populations [12,14,15,16,17,18,19,20,21,22,62,63]. Thus, fNIRS may provide a valuable tool for the assessment of cognitive function and efficacy of dual-task interventions in adults with impaired cardiovascular health.

The present study had several limitations. Firstly, as our fNIRS system had limited spatial resolution, which allowed examination only at the cortical surface of the PFC, future research is needed to examine the effects of other areas (e.g., premotor and primary motor cortex, basal ganglia, and supplementary motor area). Doing so would assist with clarifying the other cortical motor areas that are contributing to dual-task walking. Secondly, the order of presentation of the Stroop conditions (i.e., from easier to harder) may have led to potential decreases in the task effects while dual tasking. Future work should incorporate the randomization of cognitive task demands. Furthermore, as we had only 28 participants in this study, future research should incorporate more participants to verify the effects of impaired HRR on PFC activation and examine the interaction with other comorbidities to reflect the heterogeneity in older adults with cardiovascular disorders. Furthermore, as there were no other studies which defined the cut off for HRR, this study relied on a statistical cutoff, which requires further validation. Further work needs to verify these results with other dual-task paradigms. Furthermore, there was a significant difference in age between the lower and higher HRR cohorts. Although age was added as a covariate in analysis, future work should examine the effect of HRR on neural activation on a similarly aged group. Lastly, as we did not measure blood pressure in this population, future research should measure blood pressure and heart rate in this cardiovascular risk population to define differences in PFC activation among groups with different cardiovascular risk factors.

## 5. Conclusions

The present findings provide the first evidence of the effect of HRR level on PFC activation while dual-task walking. HRR is a risk factor of hypertension in older adults, and hypertension is associated with non-executive cognitive impairment in older adults. Findings from this study highlight the importance of higher heart rate reserve levels in older adults and provide a potential factor for further exploration in the examination of cardiovascular and cerebrovascular health changes in older adults. Although we did not find any regional-based differences in PFC activation among the two groups, we did find a significant relationship between HRR and PFC activation parameters. Lastly, HRR may serve as a clinical biomarker for cognitive health of an older adult with or without cardiovascular risk and, thus, merits further exploration in longitudinal studies examining the causal relationships between cardiovascular diseases and PFC activation patterns while dual-task walking in older adults. 

## Figures and Tables

**Figure 1 ijerph-19-00047-f001:**
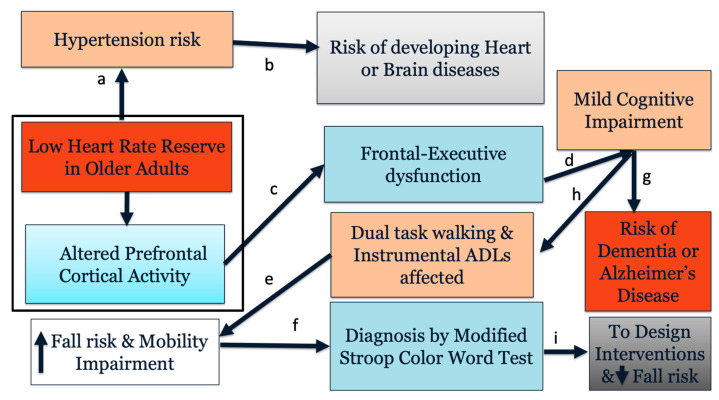
Conceptual model showing relationship between heart rate reserve and cognitive function in older adults. Box indicates associations examined by this study. Link a [7]; link b [3]; links c and e [13,25,26]; link d [27]; link f [24]; link g [28]; link h [29]; link i [30].

**Figure 2 ijerph-19-00047-f002:**
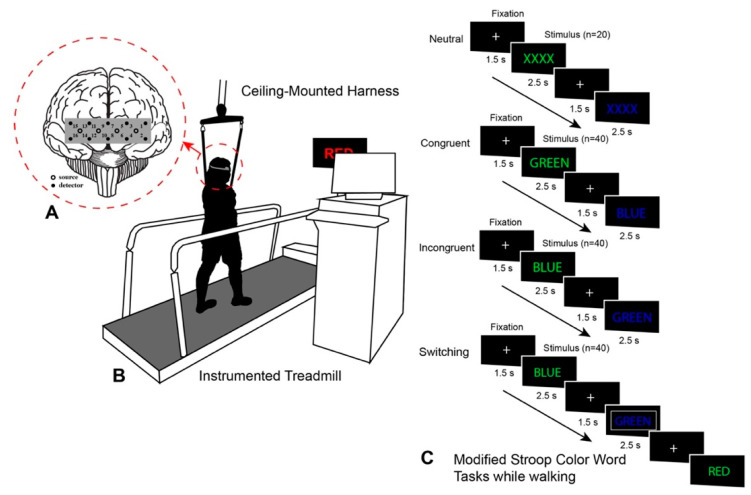
Schematic of experimental setup with fNIRS headband (**A**), instrumented treadmill and monitor with visual cues (**B**), and cues during MSCWT neutral, congruent, incongruent, and switching tasks (**C**). Note: MSCWT = Modified Stroop Color Word Test.

**Figure 3 ijerph-19-00047-f003:**
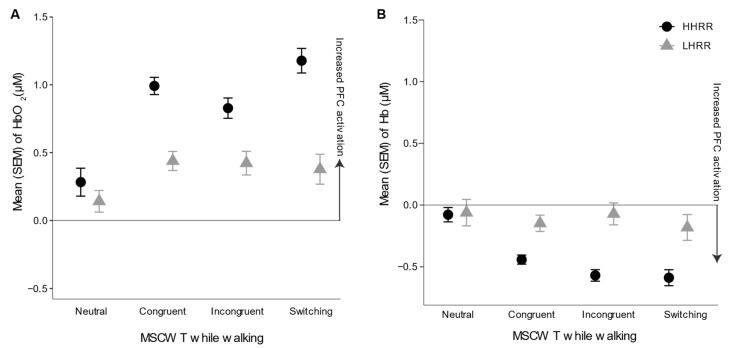
Prefrontal cortical activation differences, as measured by (**A**) mean oxygenated hemoglobin (HbO_2_) and (**B**) mean deoxygenated hemoglobin (Hb) levels across all 16 optodes between lower heart rate reserve (LHRR) and higher heart rate reserve (HHRR) groups across Modified Stroop Color Word Test (MSCWT) tasks.

**Table 1 ijerph-19-00047-t001:** Participant demographics.

Characteristic	LHRR (*n* = 14)	HHRR (*n* = 14)	*p*-Value
Males/females	4/10	6/8	0.693
Age (years)	73.8 ± 7.0	61.8 ± 8.0	0.0002 ***
Aerobic capacity (estimated VO_2_ max)	25.6 ± 8.6	33.2 ± 6.4	0.019 *
Education (years)	7.3 ± 4.5	7.5 ± 6.6	0.949
Heart rate reserve (bpm)	57.5 ± 12.5	85.3 ± 8.7	<0.0001 ***
NART	113.4 ± 6.9	110.5 ± 7.1	0.335
RBANS	96.4 ± 9.1	98.8 ± 8.6	0.498
Rockport gait speed (m/s)	1.8 ± 0.2	2.0 ± 0.3	0.030 *

Note: LHRR = lower heart rate reserve, HHRR = higher heart rate reserve, NART = National Adult Reading Test, RBANS = Repeated Battery for the Assessment of Neuropsychological Status. * *p* < 0.05, *** *p* < 0.001.

**Table 2 ijerph-19-00047-t002:** Gait speed and accuracy during MSCWT while walking.

	LHRR (*n* = 14)	HHRR (*n* = 14)
Gait speed (m/s)		
Neutral	0.97 ± 0.27	0.93 ± 0.32
Congruent	0.94 ± 0.29	1.03 ± 0.28
Incongruent	1.01 ± 0.27	1.06 ± 0.26
Switching	1.08 ± 0.27	1.07 ± 0.28
Accuracy (%)		
Neutral	99.6 ± 1.3	100.0 ± 0.0
Congruent	99.8 ± 0.7	99.8 ± 0.8
Incongruent	95.7 ± 3.1	96.3 ± 2.4
Switching ***	70.4 ± 24.0	92.8 ± 5.1

Note: LHRR = lower heart rate reserve, HHRR = higher heart rate reserve. Post hoc *t*-tests examining cohort effects at *** *p* < 0.001.

**Table 3 ijerph-19-00047-t003:** Linear mixed effect models with heart rate reserve (HRR) level and Modified Stroop Color Word Test (MSCWT) task conditions as the main effects while controlling for channel, age, and aerobic capacity and HbO_2_ as the dependent measure.

	β	SE	*p*-Value
Age	0.015	0.006	0.021 *
Aerobic capacity	0.022	0.006	<0.001 ***
Task: congruent vs. neutral	0.439	0.090	<0.001 ***
Task: incongruent vs. neutral	0.301	0.090	<0.001 ***
Task: switching vs. neutral	0.459	0.090	<0.001 ***
HRR level: HHRR vs. LHRR	0.062	0.151	0.682
Channel 12 vs. Channel 1	0.594	0.266	0.026 *
Channel 14 vs. Channel 1	0.650	0.269	0.016 *
HHRR × congruent	0.271	0.116	0.020 *
HHRR × incongruent	0.245	0.116	0.035 *
HHRR × switching	0.437	0.116	<0.001 ***

Note: Data are reported as estimates, standard error (SE), and *p*-value; LHRR = lower heart rate reserve; HHRR = higher heart rate reserve; * *p* < 0.05, *** *p* < 0.001.

**Table 4 ijerph-19-00047-t004:** Linear mixed effect models with heart rate reserve (HRR) level and Modified Stroop Color Word Test (MSCWT) task conditions as the main effects while controlling for channel, age, and aerobic capacity and Hb as the dependent measure.

	β	SE	*p* Value
Age	0.105	0.004	0.008 **
Aerobic capacity	−0.007	0.004	0.045 *
Task: congruent vs. neutral	−0.193	0.085	0.023 *
Task: incongruent vs. neutral	−0.220	0.085	<0.010 **
Task: switching vs. neutral	−0.314	0.085	<0.001 ***
HRR level: HHRR vs. LHRR	−0.216	0.105	0.039 *
Channel 8 vs. Channel 1	0.582	0.164	<0.001 ***
Channel 10 vs. Channel 1	0.422	0.164	0.010 *
Channel 12 vs. Channel 1	0.461	0.160	0.004 **
Channel 14 vs. Channel 1	0.319	0.162	<0.050 *
HHRR × congruent	−0.171	0.109	0.118
HHRR × incongruent	−0.272	0.109	0.013 *
HHRR × switching	−0.196	0.109	0.072

Note: Data are reported as estimates, standard error (SE), and *p*-value; LHRR = lower heart rate reserve; HHRR = higher heart rate reserve; * *p* < 0.05, ** *p* < 0.01, *** *p* < 0.001.

## Data Availability

The data presented in this study are available on request from the corresponding author.

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
