# Peer review of "Effect of Heart Rate Reserve on Prefrontal Cortical Activation While Dual-Task Walking in Older Adults"

_ijerph, 2021, doi:10.3390/ijerph19010047_

Round 1

Reviewer 1 Report

This is an important first study investigating the effects of heart rate reserve on prefrontal cortical activation while dual tasking in older adults.  However, a concern is that the two groups (LHRR vs. HHRR) were not matched for age; indeed, the LHRR group was 20% older than the HHRR group.  This is a major limitation and as such should be addressed in detail in the Discussion and at the least included in the limitations section.

Minor comments:

Abstract: please provide n=14 for each group.

Abstract and Methods: Please add age range so we know the definition of "older."

Author Response

We thank the reviewer for their feedback, and have revised the abstract, methods, and discussion as requested. In the abstract, we now provide the number of participants in each group and age range for the cohort. In the methods, we have also provided the age range of the cohort. In the discussion we discuss the age difference as a limitation of the study. We now state: “Further, there was a significant difference in age between the lower and higher HRR cohorts. Although age was added as a covariate in analysis, future work should examine the effect of HRR on neural activation on a similarly aged group”.

Reviewer 2 Report

Hypothesis well anchored in the postulates. Appropriate methodology and results with scientific rigor. Of all cited references, approximately 25% are most up-to-date. 

Author Response

We thank the reviewer for their positive feedback.

Reviewer 3 Report

1. Problems with grammar and expression.
2. line 63 add refernce?
3 2.2 procedures? Is this a blined study? Please fill it out additionally.
4. Can you do sample size calculation?
5. Please fill out the reliability and validity of the evaluation tool.
6. Please fill out the P value in the table.
7. Overall, there are many abbreviations that are not needed. Please organize it.

Author Response

We thank the reviewer for their feedback. We have gone through the manuscript and tried to address any observed errors in grammar and expression, particularly in the discussion. We have added the appropriate references in line 63. No, this was not a blinded study, as testing on Day 1 was required to establish the cohorts in this study, so no addition content was added in the procedures. We carried out a sample size based on earlier pilot data and now state: “A priori sample size calculations based on earlier pilot data suggested a minimum of 26 participants needed in this study to achieve 0.85 power, assuming ? = 0.05, and effect size of 0.25.” We have added a discussion of the reliability and validity of the evaluation tool in the discussion. We now state: “Overall, fNIRS provides a valid and reliable measure of prefrontal activation in adults. As demonstrated by concurrent functional magnetic resonance imaging and fNIRS studies, fNIRS measures provide a valid measure of neural activation [56-58]. Further, fNIRS measures have demonstrated good to moderate reliability in prior motor and walking tasks [59-61] and application in numerous dual task walking conditions and populations [12,14-22,62,63]. Thus, fNIRS may provide a valuable tool for the assessment of cognitive function and efficacy of dual task interventions in adults with impaired cardiovascular health.” We now have added the p-values for post-hoc t-test results in Table 2. Lastly, we have removed abbreviations used less than 2 times in the manuscript.